# Log-normality and Skewness of Estimated State/Action Values in Reinforcement Learning

**Liangpeng Zhang**[1,2], **Ke Tang**[3,1], **and Xin Yao**[3,2]
[1]School of Computer Science and Technology,
University of Science and Technology of China
[2]University of Birmingham, U.K.
[3]Shenzhen Key Lab of Computational Intelligence,
Department of Computer Science and Engineering,
Southern University of Science and Technology, China
`lxz472@cs.bham.ac.uk, tangk3@sustc.edu.cn, xiny@sustc.edu.cn`

## Abstract

Under/overestimation of state/action values are harmful for reinforcement learning agents. In this paper, we show that a state/action value estimated using the Bellman equation can be decomposed to a weighted sum of path-wise values that follow log-normal distributions. Since log-normal distributions are skewed, the distribution of estimated state/action values can also be skewed, leading to an imbalanced likelihood of under/overestimation. The degree of such imbalance can vary greatly among actions and policies within a single problem instance, making the agent prone to select actions/policies that have inferior expected return and higher likelihood of overestimation. We present a comprehensive analysis to such skewness, examine its factors and impacts through both theoretical and empirical results, and discuss the possible ways to reduce its undesirable effects.

## 1   Introduction

In reinforcement learning (RL) [1, 2], actions executed by the agent are decided by comparing relevant state values $V$ or action values $Q$. In most cases, the ground truth $V$ and $Q$ are not available to the agent, and the agent has to rely on estimated values $\hat{V}$ and $\hat{Q}$ instead. Therefore, whether or not an RL algorithm yields sufficiently accurate $\hat{V}$ and $\hat{Q}$ is a key factor to its performance. Many researches have proved that, for many popular RL algorithms such as Q-learning [3] and value iteration [4], estimated values are guaranteed to converge in the limit to their ground truth values [5, 6, 7, 8].

Still, under/overestimation of state/action values occur frequently in practice. Such phenomena are often considered as the result of insufficient sample size or the utilisation of function approximation [9]. However, recent researches have pointed out that the basic estimators of $V$ and $Q$ derived from the Bellman equation, which were considered unbiased and have been widely applied in RL algorithms, are actually biased [10] and inconsistent [11]. For example, van Hasselt [10] showed that the max operator in the Bellman equation and its transforms introduces bias to the estimated action values, resulting in overestimation. New operators and algorithms have been proposed to correct such biases [12, 13, 14], inconsistency [11] and other issues of value-based RL [15, 16, 17, 18].

This paper shows that, despite having great improvements in recent years, the value estimator of RL can still suffer from under/overestimation. Specifically, we show that the distributions of estimated state/action values are very likely to be skewed, resulting in imbalanced likelihood of under/overestimation. Such skewness and likelihood can vary dramatically among actions/policies within a single problem instance. As a result, the agent may frequently select undesirable actions/policies, regardless of its value estimator being unbiased.

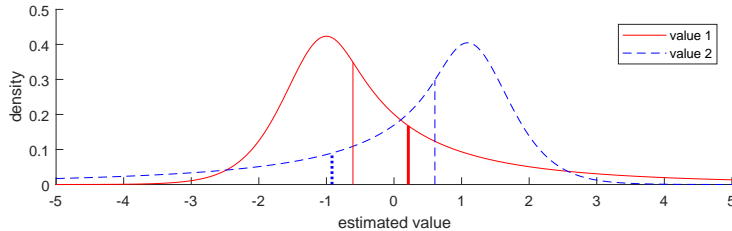

Figure 1: Illustration of positive skewness (red distribution) and negative skewness (blue distribution). Thick and thin vertical lines represent the corresponding expected values and medians, respectively.

Such phenomenon is illustrated in Figure 1. An estimated state/action value following the red distribution has a mean $0.21$ and a median $-0.61$, thus tends to be underestimated. Another following the blue distribution, on the other hand, has a mean $-0.92$ and a median $0.61$, thus likely to be overestimated. Despite that the red expected return is noticeably greater than the blue, the probability of an unbiased agent arriving at the opposite conclusion (blue is better) and thus selecting the inferior action/policy is around $0.59$, which is even worse than random guessing.

This paper also indicates that such skewness comes from the Bellman equation passing the dispersion of transition dynamics to the state/action values. Therefore, as long as a value is estimated by applying the Bellman equation to the observations of transition, it can suffer from the skewness problem, regardless of the algorithm being used. Instead of proposing new algorithms, this paper suggests two general ways to reduce the skewness. The first is to balance the impacts of positive and negative immediate rewards to the estimated values. We show that positive rewards lead to positive skewness and vice versa, and thus, a balance between the two may help neutralise the harmful effect of skewness. The second way is to simply collect more observations of transitions. However, our results in this paper indicate that the effectiveness of this approach diminishes quickly as the sample size grows, and thus is recommended only when observations are cheap to obtain.

In the rest of this paper, we will elaborate our analysis to the distributions of state/action values estimated by the Bellman equation. Specifically, we will show that an estimated value in a general MDP can be decomposed to path-wise values in normalised single-reward Markov chains. The path-wise values are shown to obey log-normal distributions, and thus the distribution of an estimated value is the convolution of such log-normal distributions. To understand which factors have the most impact to the skewness, we derive the expressions of the parameters of these log-normal distributions. We then discuss whether the skewness of estimated values can be reduced in order to improve learning performance. Finally, we provide our empirical results to complement our theoretical ones, illustrating how substantial the undesirable effect of skewness can be, as well as to what degree such effect can be reduced by obtaining more observations.

## 2   Preliminaries

The standard RL setup of [1] is followed in this paper. An environment is formulated as a finite discounted Markov Decision Process (MDP) $M = (S, A, P, R, \gamma)$, where $S$ and $A$ are finite sets of states and actions, $P(s'|s, a)$ is a transition probability function, $R(s, a, s')$ is an immediate reward function, and $\gamma \in (0, 1)$ is a discount factor. A trajectory $(s^1, a^1, s^2, r^1), (s^2, a^2, s^3, r^2), ...,$ $(s^t, a^t, s^{t+1}, r^t)$ represents the interaction history between the agent and the MDP. The number of occurrences of state-action pair $(s, a)$ and transition $(s, a, s')$ in such trajectory are denoted $N_{s,a}$ and $N_{s,a,s'}$, respectively.

A policy is denoted $\pi$, and $V^\pi(s)$ is the state value of $\pi$ starting from $s$. An action value $Q^\pi(s, a)$ is essentially a state value following a non-stationary policy that selects $a$ at the first step but follows $\pi$ thereafter. It can be analysed in the same way as $V^\pi$, so it suffices to focus on $V^\pi$ in the following sections. For convenience, superscript $\pi$ in $V^\pi$ will be dropped if it is clear from the context.

For any $s \in S$ and policy $\pi$, it holds that $V^\pi(s) = \sum_{s' \in S} P(s'|s, \pi(s))(R(s, \pi(s), s') + \gamma V^\pi(s'))$, which is called the Bellman equation. Most model-based and model-free RL algorithms utilise this equation, its equivalents, or its transforms to estimate state values. Since $P$ and $R$ are unknown to the agent, estimated values $\hat{V}(s)$ are computed from estimated transitions $\hat{P}$ and rewards $\hat{R}$ instead,

where $\hat{P}(s'|s,a) = N_{s,a,s'}/N_{s,a}$ and $\hat{R}(s,a,s') = r_t$ with $(s_t, a_t, s_{t+1}) = (s, a, s')$. This is done explicitly in model-based learning, and implicitly with frequencies of updates in model-free learning. We will show in later section that the skewness of estimated values is decided by the dynamic effects of the environment rather than the learning algorithm being used, and therefore, it suffices to focus on the model-based case in order to evaluate such skewness.

The skewness in this paper refers to the Pearson 2 coefficient $(\mathbb{E}[X] - \mathrm{median}[X])/\sqrt{\mathrm{Var}[X]}$ [19, 20]. Following this definition, a distribution has a positive skewness if and only if its mean is greater than its median, and vice versa. Assuming that the bias of $\hat{V}$ is corrected or absent, we have $\mathbb{E}[\hat{V}] = V$. Thus, a positive skewness of $\hat{V}$ means $\Pr(\hat{V} < V) > 0.5$, indicating a higher likelihood of underestimation, while a negative skewness indicates a higher likelihood of overestimation.

An informative indicator of skewness is $\mathrm{CDF}_{\hat{V}}(V) - 0.5$ where $\mathrm{CDF}_{\hat{V}}$ is the cumulative distribution function of $\hat{V}$. The sign of this indicator is consistent with the Pearson 2 coefficient, while its absolute value gives the extra probability of under/overestimation of $\hat{V}$ compared to a zero-skew distribution.

A log-normal distribution with location parameter $\mu$ and scale parameter $\sigma$ is denoted $\ln\mathcal{N}(\mu, \sigma^2)$. A random variable $X$ follows $\ln\mathcal{N}(\mu, \sigma^2)$ if and only if $\ln(X)$ follows normal distribution $\mathcal{N}(\mu, \sigma^2)$. The parameters $\mu$ and $\sigma$ of log-normal distribution can be calculated from its mean and variance by $\mu = \ln\left(\frac{\mathbb{E}[X]^2}{\sqrt{\mathbb{E}[X]^2 + \mathrm{Var}[X]}}\right)$, and $\sigma^2 = \ln\left(1 + \frac{\mathrm{Var}[X]}{\mathbb{E}[X]^2}\right)$, where $\mathbb{E}[X]$ and $\mathrm{Var}[X]$ are the mean and variance of $X \sim \ln\mathcal{N}(\mu, \sigma^2)$, respectively.

## 3 Log-normality of Estimated State Values

In this section, we elaborate our analysis to the distributions of estimated values $\hat{V}$. The analysis is formed of three steps. First, state values in general MDPs are decomposed to the state values in relevant normalised single-reward Markov chains. Second, they are further decomposed into path-wise state values. Third, the path-wise state values are shown to obey log-normal distributions.

### 3.1 Decomposing into Normalised Single-reward Markov chains

Given an MDP $M$ and a policy $\pi$, the interaction between $\pi$ and $M$ forms a Markov chain $M^\pi$, with transition probability $p_{i,j} = P(s_j|s_i, \pi(s_i))$ and reward $r_{i,j} = R(s_i, \pi(s_i), s_j)$ from arbitrary state $s_i$ to state $s_j$. Let $\boldsymbol{P}^\pi$ be the transition matrix of $M^\pi$, $\boldsymbol{V}^\pi$ be the (column) vector of state values, $\boldsymbol{R}^\pi$ be the reward matrix, and $\boldsymbol{J}$ be a vector of 1 with the same size of $\boldsymbol{V}^\pi$. Then Bellman equation is equivalent to $\boldsymbol{V}^\pi = \boldsymbol{P}^\pi \circ \boldsymbol{R^\pi} \boldsymbol{J} + \gamma \boldsymbol{P}^\pi \boldsymbol{V}^\pi = (\boldsymbol{I} - \gamma \boldsymbol{P}^\pi)^{-1}(\boldsymbol{P}^\pi \circ \boldsymbol{R^\pi} \boldsymbol{J})$, where $\boldsymbol{I}$ is an identity matrix, and $\circ$ is Hadamard product.

This equation indicates that a state value is a weighted sum of *dynamic effects*, with rewards serving as the weights of summation. Precisely, let $\boldsymbol{B} = (\boldsymbol{I} - \gamma \boldsymbol{P}^\pi)^{-1}$, then the equation above becomes $\boldsymbol{V}^\pi = \boldsymbol{B}(\boldsymbol{P}^\pi \circ \boldsymbol{R^\pi} \boldsymbol{J})$, or $V^\pi(s_i) = \sum_{j,k} r_{j,k}(b_{i,j}\, p_{j,k})$. Here, term $(b_{i,j}\, p_{j,k})$ describes the joint dynamic effect starting from $s_i$ ending with transition $s_j s_k$, which will be elaborated in Section 3.2.

Let $M_{j,k}^\pi$ denote a *normalised single-reward Markov chain* (NSR-MC) of $M^\pi$, which has exactly the same $S$, $A$, $\gamma$ and $P^\pi$ as $M^\pi$, but all rewards are trivially 0 except $r_{j,k} = 1$. For an NSR-MC $M_{j,k}^\pi$, the equation above becomes $V_{M_{j,k}^\pi}^\pi(s_i) = b_{i,j}\, p_{j,k}$. Thus, a state value $V$ of a general MDP $M$ can be rewritten as the weighted sum of state values of all $|S|^2$ NSR-MCs $\{M_{j,k}^\pi\}$ of $M$, i.e.

$$V_M^\pi(s_i) = \sum_{j,k} r_{j,k} V_{M_{j,k}^\pi}(s_i). \tag{1}$$

Therefore, the next step of analysis is to examine the state values in NSR-MCs.

### 3.2 Decomposing into Path-wise State Values

Seeing Markov chain $M^\pi$ as a directed graph, a *walk* $w$ of length $|w|$ in such graph is a sequence of $|w|$ successive transitions through states $s^1, s^2, s^3, ..., s^{|w|+1}$.[1] A *path* is a walk without repeated states, with exception to the last state $s^{|w|+1}$, which can be either a visited or an unvisited one.

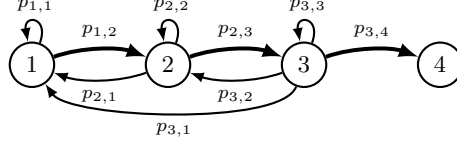

Figure 2: Illustration of walks and a representative path. "Forward" and "backward" transitions are drawn in thick and thin arrows, respectively, and $p_{i,j}$ denotes the transition probability from $s_i$ to $s_j$.

In an NSR-MC with unique non-zero reward $r_{j,k} = 1$, a state value $V^\pi(s_i) = b_{i,j}\,p_{j,k}$ can be expanded as a sum of the discounted occurrence probabilities of walks that start from $s_i$ and end with transition $(s_j, \pi(s_j), s_k)$. Let $W_{i,j,k}$ denotes the set of all possible walks $w$ satisfying $s^1 = s_i$, $s^{|w|} = s_j$ and $s^{|w|+1} = s_k$. Then we have $V(s_i) = \sum_{w \in W_{i,j,k}} (\gamma^{|w|-1} \prod_{(s^t,s^{t+1}) \text{ on } w} p_{s^t, s^{t+1}})$. Since $W_{i,j,k}$ is infinite, the walks in $W_{i,j,k}$ need to be put into finite groups for further analysis.

Concretely, a step in a walk is considered "forward" if it arrives to a previously unvisited state, and "backward" if the destination has already been visited before that step. The latter also includes the cases where $s^{t+1} = s^t$, that is, the agent stays at the same state after transition. The only exception to this classification is the last transition of a walk, which is always considered a "forward" one, regardless of if its destination having been visited or not. The start state $s^1$ and all such "forward" transitions of a walk $w$ form a *representative path* of $w$, denoted $\tilde{w}$.

This is illustrated by Figure 2. In this example, all walks from $s_1$ passing $s_2$ ending with $s_3 s_4$, such as $(s_1 s_1 s_2 s_3 s_3 s_4)$, $(s_1 s_2 s_3 s_1 s_2 s_3 s_4)$ and $(s_1 s_2 s_3 s_2 s_3 s_2 s_3 s_4)$, are grouped with the representative path $(s_1 s_2 s_3 s_4)$. Note that transition $s_1 s_3$ will not happen within this group; rather, it belongs to the groups that have $s_1 s_3$ in their representative paths.

As can be seen from Figure 2, all possible walks sharing one representative path $\tilde{w}$ compose a chain which has the same transition probability values with the original Markov chain $M^\pi$, but with only two type of transitions: (forward) $s^i$ to $s^{i+1}$ ($i \le |\tilde{w}|$); (backward) $s^i$ to $s^j$ ($j \le i \le |\tilde{w}|$). We call this chain the *derived chain* of $\tilde{w}$, denoted $M^\pi(\tilde{w})$, or simply $M(\tilde{w})$. Then the infinite sum becomes

$$V(s) = \sum_{\tilde{w} \in \tilde{W}} V_{M(\tilde{w})}(s), \tag{2}$$

where $\tilde{W}$ is the set of all representative paths that start from $s$ and end with the unique 1-reward transition of the relevant NSR-MC. Such $V_{M(\tilde{w})}(s)$ are called *path-wise state values* of $M^\pi$.

Since the main concern of this paper is the skewness of $\hat{V}$, we do not provide a constructive method of obtaining all $M^\pi(\tilde{w})$. Rather, we point out that the size of $\tilde{W}$ is at most $(|S|!)$, and thus an estimated value $\hat{V}$ in NSR-MCs can be decomposed to finitely many estimated path-wise state values.

### 3.3 Log-normality of Estimated Path-wise State Values

Strictly speaking, derived chain $M(\tilde{w})$ of a representative path $\tilde{w}$ is not necessarily a Markov chain, because only part of the transitions in the original Markov chain $M^\pi$ is included, allowing the possibility of $\sum_{j=1}^{i+1} p_{s^i,s^j} < 1$. However, this does not make the path-wise state values violate Bellman equation, and thus they can be treated as regular state values.

Since a representative path $\tilde{w}$ has no repeated states (except for $s^{|\tilde{w}|+1}$ which can either be a new state or the same as some $s^k$), the superscripts here can be treated as the indices of states for convenience. Therefore, path-wise state value $V_{M(\tilde{w})}(s^i)$ is denoted $V_i$, and $p_{i,j}$ refers to $p_{s^i,s^j}$ in this section. Given $\tilde{w}$, the most important path-wise value is $V_1$, which belongs to the start point of $\tilde{w}$.

**Definition 3.1.** Given a derived chain $M(\tilde{w})$ and discount factor $\gamma$, let $p_{i,j}$ be the transition probability from $s^i$ to $s^j$ on $M(\tilde{w})$. The *joint dynamic effect* of $M(\tilde{w})$ for $i \le |\tilde{w}|$ is recursively defined as

$$D_i = \frac{\gamma p_{i,i+1}}{1 - \gamma(p_{i,i} + \sum_{j=1}^{i-1} p_{i,j} \prod_{k=j}^{i-1} D_k)}.$$

**Lemma 3.2.** *For all $i < |\tilde{w}|$, path-wise state values satisfy $V_i = D_i V_{i+1}$.*

*Proof.* By Bellman equation, it holds that $V_i = \sum_{j=1}^{|\tilde{w}|+1} p_{i,j}(r_{i,j}+\gamma V_j)$. By definition of $M(\tilde{w})$ we have $p_{i,j} = 0$ for $j > i+1$ and $r_{i,j} = 0$ for $(i,j) \neq (|\tilde{w}|,|\tilde{w}|+1)$. Thus $V_i = \gamma \sum_{j=1}^{i+1} p_{i,j}V_j$ for $i < |\tilde{w}|$. When $i = 1$, this becomes $V_1 = \gamma(p_{1,1}V_1+p_{1,2}V_2) = \frac{\gamma p_{1,2}}{1-\gamma p_{1,1}}V_2 = D_1 V_2$. Suppose $V_i = D_i V_{i+1}$ holds for all $i \leq k < |\tilde{w}|-1$. Then $V_i = (\prod_{j=i}^{k} D_j)V_{k+1}$ for $i \leq k$, and therefore, $V_{k+1} = \gamma \sum_{j=1}^{k+2} p_{k+1,j}V_j = \gamma[\sum_{j=1}^{k+1} p_{k+1,j}(\prod_{l=j}^{k} D_l)V_{k+1} + p_{k+1,k+2}V_{k+2}] = \frac{\gamma p_{k+1,k+2}}{1-\gamma(p_{k+1,k+1}+\sum_{j=1}^{k} p_{k+1,j} \prod_{l=j}^{k} D_l)}V_{k+2} = D_{k+1}V_{k+2}$. Thus, by the principle of induction, $V_i = D_i V_{i+1}$ holds for all $i < |\tilde{w}|$. $\square$

**Lemma 3.3.** *For all $i \leq |\tilde{w}|$, $V_i = \frac{1}{\gamma} \prod_{j=i}^{|\tilde{w}|} D_j$. Particularly, $V_1 = \frac{1}{\gamma} \prod_{j=1}^{|\tilde{w}|} D_j$.*

*Proof.* By definition of $\tilde{w}$, there are two possible cases of the last step from $s^{|\tilde{w}|}$ to $s^{|\tilde{w}|+1}$: (I) $s^{|\tilde{w}|+1} \notin \{s^1, ..., s^{|\tilde{w}|}\}$; (II) there exists $k \leq |\tilde{w}|$ such that $s^{|\tilde{w}|+1} = s^k$.

(Case I) There is no transition starting from $s^{|\tilde{w}|+1}$ in this case, thus $V_{|\tilde{w}|+1} = 0$. Therefore, $V_{|\tilde{w}|} = p_{|\tilde{w}|,|\tilde{w}|+1}(r_{|\tilde{w}|,|\tilde{w}|+1} + \gamma V_{|\tilde{w}|+1}) + \gamma \sum_{j=1}^{|\tilde{w}|} p_{|\tilde{w}|,j}V_j = p_{|\tilde{w}|,|\tilde{w}|+1} + \gamma \sum_{j=1}^{|\tilde{w}|} p_{|\tilde{w}|,j}V_j = \frac{p_{|\tilde{w}|,|\tilde{w}|+1}}{1-\gamma(p_{|\tilde{w}|,|\tilde{w}|}+\sum_{j=1}^{|\tilde{w}|-1} p_{|\tilde{w}|,j} \prod_{k=j}^{|\tilde{w}|-1} D_k)} = \frac{1}{\gamma}D_{|\tilde{w}|}$. Thus $V_i = (\prod_{j=i}^{|\tilde{w}|-1} D_j)V_{|\tilde{w}|} = \frac{1}{\gamma} \prod_{j=i}^{|\tilde{w}|} D_j$.

(Case II with $s^{|\tilde{w}|+1} = s^k$) In this case $V_{|\tilde{w}|+1} = V_k$ and $p_{|\tilde{w}|,|\tilde{w}|+1} = p_{|\tilde{w}|,k}$, thus $V_{|\tilde{w}|} = p_{|\tilde{w}|,|\tilde{w}|+1}(r_{|\tilde{w}|,|\tilde{w}|+1} + \gamma V_k) + \gamma \sum_{j=1,j\neq k}^{|\tilde{w}|} p_{|\tilde{w}|,j}V_j = p_{|\tilde{w}|,|\tilde{w}|+1} + \gamma \sum_{j=1}^{|\tilde{w}|} p_{|\tilde{w}|,j}V_j$ which is the same expression as the first case, and therefore $V_i = \frac{1}{\gamma} \prod_{j=i}^{|\tilde{w}|} D_j$ also holds for this case. $\square$

In both of the two cases above, $V_1$ is the product of $D_1, D_2, ..., D_{|\tilde{w}|}$ given by Definition 3.1, and an additional factor $\frac{1}{\gamma}$. Thus we have $\ln(V_1) = -\ln(\gamma) + \sum_{j=1}^{|\tilde{w}|} \ln(D_j)$. By replacing all $p_{i,j}$ in Definition 3.1 with estimated transition $\hat{p}_{i,j}$, we get the "estimated" [2] joint dynamic effects $\hat{D}$. Then the equation above becomes $\ln(\hat{V}_1) = -\ln(\gamma) + \sum_{j=1}^{|\tilde{w}|} \ln(\hat{D}_j)$. Assuming $\hat{D}_i$'s as independent random variables, it can be shown by the central limit theorem that as $|\tilde{w}|$ grows, $\ln(\hat{V}_1)$ will tend to a normal distribution, and therefore, $\hat{V}_1$ approximates a log-normal distribution.

The "estimated" joint dynamic effects $\hat{D}$ are actually mutually dependent in most cases, thus the rigorous analysis of log-normality is more complicated. The main idea here is to first prove all $\hat{D}_i \leq \gamma$, and then show that the summation involving terms $p_{i,j} \prod_{k=j}^{i-1} \hat{D}_k$ in Definition 3.1 diminish quickly with the size of $\tilde{w}$, which indicates that $\hat{D}_i$ is mostly decided by $\hat{p}_{i,i}$ and $\hat{p}_{i,i+1}$ and thus the dependency between any two $\hat{D}$ is relatively weak. As the focus here is to see the skewness of $\hat{V}_1$, such analysis is skipped, and we proceed to the study of parameters of log-normal distribution of $\hat{V}_1$.

Since $\hat{p}_{i,i}$ and $\hat{p}_{i,i+1}$ are the main factors that decide $\hat{D}_i$, we provide the result on the most representative case where $p_{i,i} + p_{i,i+1} = 1$ and all other $p_{i,j}$ are 0 for $i < |\tilde{w}|$. Such $M(\tilde{w})$ is denoted $M_0(\tilde{w})$ in the following text. It is easy to see that all $\hat{D}_i$ are mutually independent in such chains.

The delta method [21, 22] below is used to obtain the expressions of parameters.

**Lemma 3.4** (Delta method[21, 22]). *Suppose $X$ is a random variable with finite moments, $\mathbb{E}[X]$ being its mean and $\mathrm{Var}[X]$ being its variance. Suppose $f$ is a sufficiently differentiable function. Then it holds that $\mathbb{E}[f(X)] \approx f(\mathbb{E}[X])$, and $\mathrm{Var}[f(X)] \approx f'(\mathbb{E}[X])^2 \mathrm{Var}[X]$.*

**Lemma 3.5.** *Let $\hat{D}_j$ be $D_j$ replacing all $p$ with $\hat{p}$. Let $N_i$ denotes the number of visits to the chain state $s^i$ in a learning trajectory. In $M_0(\tilde{w})$ derived chains it holds that $\mathbb{E}[\hat{D}_j] \approx \frac{\gamma p_{j,j+1}}{1-\gamma p_{j,j}}$, and $\mathrm{Var}[\hat{D}_j] \approx \frac{\gamma^2(1-\gamma)^2}{(1-\gamma p_{j,j})^4} \cdot \frac{p_{j,j}p_{j,j+1}}{N_j}$.*

*Proof.* It holds that $\mathrm{Var}[\hat{p}_{j,j+1}] = (\frac{1}{N_j})^2 N_j p_{j,j} p_{j,j+1} = \frac{p_{j,j}p_{j,j+1}}{N_j}$, then by applying Lemma 3.4 to Definition 3.1. $\square$

**Lemma 3.6.** *In $M_0(\tilde{w})$ derived chains it holds that*

$$\mathbb{E}[\hat{V}_1] = \frac{1}{\gamma} \prod_{j=1}^{|\tilde{w}|} \mathbb{E}[\hat{D}_j],$$

$$\text{Var}[\hat{V}_1] \approx \frac{1}{\gamma^2} \left( \prod_{j=1}^{|\tilde{w}|} (\text{Var}[\hat{D}_j] + \mathbb{E}[\hat{D}_j]^2) - \prod_{j=1}^{|\tilde{w}|} \mathbb{E}[\hat{D}_j]^2 \right).$$

*Proof.* For independent $X_1, X_2, ..., X_n$ it holds that $\text{Var}[X_1...X_n] = \prod_{j=1}^{n} (\text{Var}[X_j] + \mathbb{E}[X_j]^2) - \prod_{j=1}^{n} \mathbb{E}[X_j]^2$. Since all $\hat{D}$ are independent in $M_0(\tilde{w})$, by applying this and Lemma 3.4 to Lemma 3.3, the above results can be obtained. $\square$

**Theorem 3.7.** *In $M_0(\tilde{w})$ with sufficiently large $|\tilde{w}|$, it holds that $\hat{V}_1 \overset{\cdot}{\sim} \ln\mathcal{N}(\mu, \sigma^2)$ with $\mu = \ln\left(\frac{\mathbb{E}[\hat{V}_1]^2}{\sqrt{\mathbb{E}[\hat{V}_1]^2 + \text{Var}[\hat{V}_1]}}\right)$ and $\sigma^2 = \ln\left(1 + \frac{\text{Var}[\hat{V}_1]}{\mathbb{E}[\hat{V}_1]^2}\right)$, where $\mathbb{E}[\hat{V}_1]$ and $\text{Var}[\hat{V}_1]$ are given by Lemma 3.6.*

*Proof.* By applying the equations on the parameters of log-normal (see Section 2) to $\hat{V}_1$. $\square$

## 4  Skewness of Estimated State Values, and Countermeasures

This section interprets the results presented in Section 3 in terms of skewness, and discuss how to reduce the undesirable effects of skewness. The skewness is mainly decided by two factors: (a) parameter $\sigma$ of log-normal distributions; (b) non-zero immediate rewards.

### 4.1  Impact of Parameter $\sigma$ of Log-normal Distributions

A regular log-normal distribution $\ln\mathcal{N}(\mu, \sigma^2)$ has a positive skewness, which means a sampled value from such distribution has more than 0.5 probability to be less than its expected value, resulting in a higher likelihood of *underestimation*. Precisely, if $X \sim \ln\mathcal{N}(\mu, \sigma^2)$, then $\mathbb{E}[X] = \exp(\mu + \sigma^2/2)$ and median$[X] = \exp(\mu)$, thus the Pearson 2 coefficient of $X$ is greater than 0. Additionally, since $\ln\mathcal{N}(\mu, \sigma^2)$ has a CDF$(x) = 0.5(1 + \text{erf}(\frac{\ln(x)-\mu}{\sqrt{2}\sigma}))$ where erf$(x)$ is the Gauss error function, our indicator CDF$(\mathbb{E}[X]) - 0.5$ equals to $0.5\,\text{erf}(\sigma/\sqrt{8})$. This indicates that $\sigma$ has a stronger impact than $\mu$ to the scale of the skewness in log-normal distributions.

Combining Lemma 3.6 and Theorem 3.7 shows that $\sigma$ is decided by a complicated interaction between all observed dynamic effect $\hat{D}_j$'s. By Lemma 3.5, transition probabilities $p_{j,*}$ completely decide $\mathbb{E}[\hat{D}_j]$, and have substantial impacts to $\text{Var}[\hat{D}_j]$.

This indicates that the main cause of skewness is the transition dynamics of MDPs rather than learning algorithms. As an extreme case, if the forward transition of a state-action pair is deterministic (i.e. $p_{j,j+1} = 1$), then its $\text{Var}[\hat{D}_j] = 0$, resulting no contribution to the skewness. If an estimated value consists of a large portion of such transitions, then the likelihoods of overestimation and underestimation are both very low. On the other hand, if backward transition probability $p_{j,j}$ (or any $p_{j,k}$ with $k \leq j$) is close to 1, then $\text{Var}[\hat{D}_j]$ increases dramatically, resulting a noticeable skewness. Real-world problems can be a mix of these two extremes, which leads to a great variety of skewness among different actions/policies, making learning significantly more difficult.

By Lemma 3.5, $\sigma$ is also dependent to the number of observations $N_j$. As $N_j$ grows infinitely, $\text{Var}[\hat{D}_j]$ slowly decreases to 0, which reduces $\text{Var}[\hat{V}_1]$ in Lemma 3.6 and eventually leads $\sigma$ to 0. This indicates that running algorithms more steps does help reduce the skewness of estimated values and improve the overall performance. However, the expression of $\text{Var}[\hat{D}_j]$ in Lemma 3.5 also indicates that the degree of improvement diminishes quickly as $N_j$ grows. Therefore, collecting more observations is not always an efficient way to reduce the skewness.

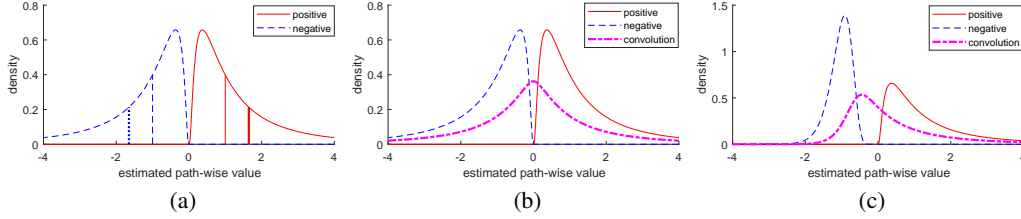

Figure 3: (a) Log-normals weighted by positive reward (red) and negative reward (blue). Thick/thin vertical lines are means & medians. (b, c) Convolution of two log-normals, given by the purple curve.

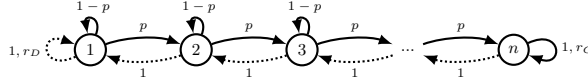

Figure 4: A chain MDP with $n$ states, forward probability $p$, goal reward $r_G$ and distraction reward $r_D$. Transitions under taking action $a^+$ is drawn in solid arrows, and $a^-$ in dotted arrows.

## 4.2 Impact of Non-zero Immediate Rewards

Non-zero immediate rewards decide not only the scale of skewness, but also the *direction* of skewness. By Equation 1 and 2 in Sections 3.1 and 3.2, path-wise values are weighted by their corresponding immediate rewards before being summed into state values. If a path-wise state value is weighted by a positive reward, then the resulting distribution is still a regular log-normal, which has a positive skewness and thus a higher likelihood of underestimation. However, if it is weighted by a negative reward, then the result is a flipped log-normal, which has a negative skewness and thus a higher likelihood of *overestimation*. This is illustrated in Figure 3 (a), where the red and blue distributions correspond to the estimated path-wise values weighted by a positive and a negative reward, respectively.

In general cases, the sum of positively skewed random variables is not necessarily a positively skewed random variable. However, the sum of regular log-normal random variables can be approximated by another log-normal [23], thus is still positively skewed. Since path-wise state values are approximately log-normal, it is clear that if an MDP only has positive immediate rewards, then all estimated values are likely to be positively skewed and thus have higher likelihoods to be underestimated.

On the other hand, if an estimated value is composed of both positive and negative rewards, then the skewness of regular and flipped log-normal distributions may partly be neutralised in their convolution. The purple distribution in Figure 3 (b) shows the result of convolution of two skewed distributions that lie symmetrically to $x = 0$. The skewness is perfectly neutralised in this case, resulting in a symmetric distribution with a balanced likelihood of under/overestimation. In the case of Figure 3 (c), the convolution is still a skewed one, but the scale of this skewness is less than the original ones.

To make learning easier, one may hope to design the reward function such that the more desirable actions/policies have both higher expected returns and higher likelihood of overestimation than the less desirable ones. However, the former requires more positive rewards, while the latter calls for more negative rewards, causing an unsolvable dilemma. Therefore, it is more realistic just to balance the likelihood of under/overestimation, so that all actions/policies can compete fairly with each other. Reward shaping [24, 25] can be a promising choice to achieve this goal, as it preserves the optimality of policies. Since a better balance of positive and negative rewards directly reduces the impact of the skewness of all relevant log-normal distributions, this approach might be more effective than simply collecting more observations.

## 5 Experiments

In this section, we present our empirical results on the skewness of estimated values. There are two purposes in these experiments: (a) to demonstrate how substantial the harm of the skewness can be; (b) to see the improvement provided by collecting more observations, as mentioned in Section 4.1.

We conducted experiments in chain MDPs shown in Figure 4. There are $n > 0$ states $s_1, s_2, ..., s_n$ in a chain MDP. At each state, the agent has two possible actions $a^+$ and $a^-$. By taking $a^+$ at $s_i$ with

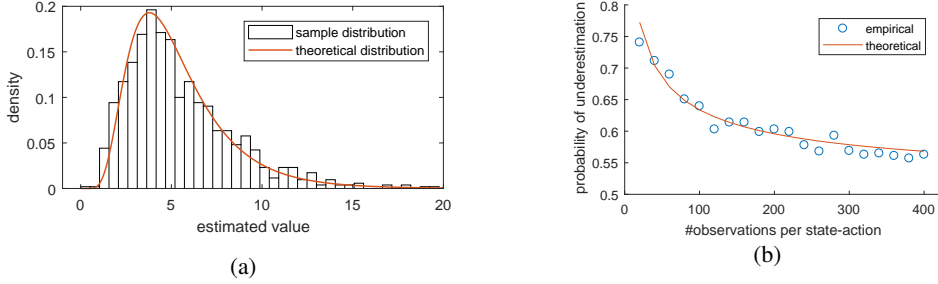

Figure 5: (a) Distribution of $\hat{V}^{\pi^+}(s_1)$ at $m = 200$. (b) Underestimation probability curve.

$i < n$, the agent has probability $p > 0$ to be sent to $s_{i+1}$, and $1 - p$ to remain at $s_i$. Taking $a^+$ at $s_n$ yields a goal reward $r_G > 0$, and the agent remains at $s_n$. Taking $a^-$, on the other hand, sends the agent from $s_i$ to $s_{i-1}$ ($i > 1$) or $s_1$ ($i = 1$) with probability 1, and if $a^-$ is taken at $s_1$, then the agent will be provided a distraction reward $r_D > 0$.

The objective of the learning agent is to discover a policy that leads it to the goal $s_n$ and collects $r_G$ as often as possible, rather than being distracted by $r_D$. There are two policy of interest: $\pi^+$ that always take $a^+$, and $\pi^-$ that always take $a^-$. Other policies can be proved to be always worse than $\pi^+$ and $\pi^-$ in terms of $V^\pi(s_1)$ regardless of $r_G$, $r_D$, $p$, and discount factor $\gamma$.

Since using max operator may introduce bias [10], we modified the default value iteration algorithm [4] to let it output the unbiased estimated state values by following predetermined policies rather than using max operator. In each run of experiment, $m$ observations were collected for each state-action pair, resulting in a data set of size $2mn$. Then, the observations were passed to the modified value iteration algorithm to estimate the state values of $\pi^+$ and $\pi^-$ under discount factor $\gamma = 0.9$.

The Markov chain $M^{\pi^+}$ and $M^{\pi^-}$ here are both single-path ones, and thus the corresponding theoretical distributions of $\hat{V}$ can be computed directly by applying Theorem 3.7. Further, since transition probabilities in $M^{\pi^-}$ are all 1, we have $\mathrm{Var}[\hat{V}^{\pi^-}] = 0$, and thus its estimated values always equal trivially to the ground truth one (i.e. it will never be under/overestimated).

The empirical and theoretical distributions of estimated state value $\hat{V}^{\pi^+}(s_1)$ with $m = 200$, $n = 20$, $p = 0.1$, $r_G = $ 1e6 in 1000 runs is shown in Figure 5 (a). One-sample Kolmogorov-Smirnov test was conducted against the null hypotheses that the empirical data came from the theoretical log-normal distributions. The resulting p-value was 0.1190, which failed to reject the null hypothesis at 5% significance level, indicating no significant difference between the theoretical and sample distribution.

More importantly, Figure 5 (a) shows a clear positive skewness, indicating a higher likelihood of underestimation. The empirical value of indicator $\mathrm{CDF}(\mathbb{E}[\hat{V}]) - 0.5$ was $+0.103$, meaning that in 60.3% of runs, the state value was underestimated. This further indicates that, if the distraction reward $r_D$ is set to a value such that $V^{\pi^-}(s_1)$ is slightly less than $V^{\pi^+}(s_1)$, then the agent will wrongly select $\pi^-$ with probability close to 0.603, which is worse than random guess.

To see whether collecting more observations helps reduce skewness, the same experiments as above were conducted with the number of observations per state-action $m$ ranged from 20 to 400. Figure 5 (b) shows the theoretical and empirical probability of underestimation $\mathrm{Pr}(\hat{V}^{\pi^+}(s_1) < \mathbb{E}\hat{V}^{\pi^+}(s_1))$. At $m = 20$, 200 and 400, the empirical underestimate probability was 0.741, 0.603 and 0.563, respectively. While from $m = 20$ to 200 there was an significant improvement of 0.138, or a 18.6% relative improvement, from 200 to 400 it was only 0.040, or 6.6% relative. This result supports the analysis in Section 4.1, demonstrating that the merit of collecting more observations is most noticeable when the sample size is low, and diminishes quickly as the sample size grows.

We also conducted experiments in the complex maze domain [26] in the same manner as above. In this domain, the task of the agent is to find a policy that can collect all flags and bring them to the goal as often as possible, without falling into any traps. The maze used is given in Figure 6 (a).

The states in this domain is represented by the current position of the agent and the status of the three flags. The agent starts at the start point indicated by S with no flag. At each time step, the agent can

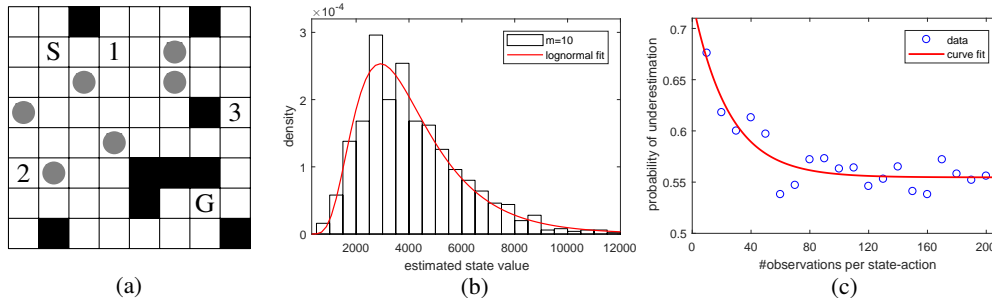

Figure 6: (a) A complex maze. S, G, numbers, and circles stand for start, goal, flags, and traps, respectively. (b) Distribution of $\hat{V}^{\pi^*}(s_{\text{start}})$ at $m = 10$. (c) Underestimation probability curve.

select one of the four directions to move to. The agent is then sent to the adjacent grid at the chosen direction with probability 0.7, and at each of the other three directions with probability 0.1, unless the destination is blocked, in which case the agent remains at the current grid. Additionally, at the flag grids (numbers in Figure 6 (a)), taking actions also provides the corresponding flag to the agent if that flag has not been obtained yet. At the goal point (G), taking arbitrary action yields an immediate reward equals to 1, 100, $100^2$ or $100^3$ if the agent holds 0, 1, 2 or 3 flags, respectively. Then the agent is sent back to the start point, and all three flag are reset to their initial position. Finally, at any trap grid (circles), taking actions sends the agent to S and resets all flags without yielding a goal reward.

The complex maze in Figure 6(a) has 440 states, 4 actions, 32 non-zero immediate rewards, and complicated transition patterns, and thus is difficult to analyse manually. However, it is noticeable that all non-zero immediate rewards are positive, and thus according to Section 4.2, estimated state values are likely to have positive skew, resulting in greater likelihood of underestimation.

Figure 6 (b) shows the empirical distribution of estimated value $\hat{V}^{\pi^*}(s_{\text{start, no flag}})$ under $\gamma = 0.9$ and $m = 10$ in 1000 runs. Although it is not a path-wise state value, the distribution is approximately log-normal with parameter $\mu \approx 8.21, \sigma \approx 0.480$. In 67.6% of these 1000 runs, the optimal state value at the start state was underestimated.

The effect of collecting a larger sample is show in Figure 6 (c). The probability of underestimation decreased from 0.676 at $m = 10$ to 0.597 at $m = 50$, 0.563 at $m = 100$, and 0.556 at $m = 200$. The data points approximated an exponential function $y = 0.1725 \exp(-0.04015x) + 0.5546$, which suggests that it can be very difficult to achieve underestimation probability lower than 0.55 by collecting more data in this domain.

## 6   Conclusion and Future Work

This paper has shown that estimated state values computed using the Bellman equation can be decomposed to the relevant path-wise state values, and the latter obey log-normal distributions. Since log-normal distributions are skewed, the estimated state values also have skewed distributions, resulting in imbalanced likelihood of under/overestimation, which can be harmful for learning.

We have also pointed out that the direction of such imbalance is decided by the immediate reward associated to the log-normal distributions, and thus, by carefully balancing the impact of positive and negative rewards when designing the MDPs, such undesirable imbalance can possibly be neutralised. Collecting more observations, on the other hand, helps reduce the skewness to a degree, but such effect becomes less significant when the sample size is already large.

It would be interesting to see how the skewness studied in this paper interacts with function approximation (e.g. neural networks [27, 28]), policy gradient [29, 30], or Monte-Carlo tree search [31, 32]. A reasonable guess is that these techniques introduce their own skewness, and the two different skewness amplify each other, making learning even more difficult. On the other hand, reducing the skewness discussed in this paper may improve learning performance even when such techniques are used. Therefore, developing a concrete method of balancing positive and negative rewards (as discussed in Section 4.2) can be very helpful, and will be investigated in the future.

**Acknowledgements**

This paper was supported by Ministry of Science and Technology of China (Grant No. 2017YFB1003102), the National Natural Science Foundation of China (Grant Nos. 61672478 and 61329302), the Science and Technology Innovation Committee Foundation of Shenzhen (Grant No. ZDSYS201703031748284), EPSRC (Grant No. J017515/1), and in part by the Royal Society Newton Advanced Fellowship (Reference No. NA150123).

## Footnotes

[1] Superscripts here refer to the timestamps on $w$ rather than the indices of specific states in $S$.

[2]Such "estimation" is not done explicitly in actual algorithms, but implicitly when using Bellman equation.

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
