[Reviews · NeurIPS 2017]

Reviewer 1



I found this to be a very interesting paper, with a nice analysis of an underappreciated property of value functions in reinforcement learning. Potential points of improvement include the following: - The theory is quite specific, and it would be nice if this could be generalized (e.g., not focusing as much on NSR-MCs). - The experiments are very small and contrived. I'm okay with this, for this paper, but it would of course be interesting to test these ideas in more interesting problems. For instance, could we measure the skewness of values in a problem of interest? And if we correct for this, does this help at scale? Even so, I'm okay with accepting the paper as is and leaving that for future work (potentially by other researchers).

Reviewer 2



This paper focuses on the problem arising from skewness in the distribution of value estimates, which may result in over- or under-estimation. With careful analysis, the paper shows that a particular model-based value estimate is approximately log-normally distributed, which is skewed and thus leading to the possibility of over- or under-estimation. It is further shown that positive and negative rewards induce opposite sort of skewness. With simple experiments, the problem of over/underestimation is illustrated. This is an interesting paper with some interesting insights on over/underestimation of values. I would like the reviewers to clarify the following two issues. The results are implied for a particular kind of model-based value estimation method, more specifically, one which relies on definition 3.1. The implication of this result to other kinds of estimation methods, especially the model-free ones is not clear. A discussion is required on why a similar effect is expected with other kinds of values estimation method. It is disappointing to have no experiments with the suggested remedies, such as the balancing the impact of positive and negative rewards and reward shaping. Some experiments are expected here to see whether these are indeed helpful. And an elaborate discussion of reward balancing and how to achieve it could also improve the quality of the paper.

Reviewer 3



This paper approaches an important and fundamental problem in RL from a new direction. Overall, I found the introductory material did an excellent job of highlighting its importance and that the main results were straightforward to follow. I did feel a bit like I lost some of the grounding intuition during section 3.3. There is just the single primary contribution of the paper, but it is well done and I think stands on its own well enough to be accepted. The experimental section felt a bit like an afterthought. It does add to the paper, but the authors could have gotten much more out of this section by showing that the log-normality *really* does show up in practice in a larger variety of MDPs. Essentially I would argue for the same sort of experiment, but including more simple MDPs and maybe some non-chain classic RL domain as well. The work has some interesting potential connections with other topics in RL, although the authors do not make these connections explicit. The distributions considered in the paper are also the focus for those interested in parameter uncertainty, posterior sampling, and exploration.